# Differential impact of self and environmental antigens on the ontogeny and maintenance of CD4[+] T cell memory

Thea Hogan[1†], Maria Nowicka[2†], Daniel Cownden[3], Claire F Pearson[4], Andrew J Yates[2‡*], Benedict Seddon[1‡*]

[1]Institute of Immunity and Transplantation, Division of Infection and Immunity, University College London, London, United Kingdom; [2]Department of Pathology and Cell Biology, Columbia University Medical Center, New York, United States; [3]Institute of Infection, Immunity and Inflammation, University of Glasgow, Glasgow, United Kingdom; [4]Kennedy Institute of Rheumatology, University of Oxford, Oxford, United Kingdom

**Abstract** Laboratory mice develop populations of circulating memory CD4[+] T cells in the absence of overt infection. We have previously shown that these populations are replenished from naive precursors at high levels throughout life (Gossel et al., 2017). However, the nature, relative importance and timing of the forces generating these cells remain unclear. Here, we tracked the generation of memory CD4[+] T cell subsets in mice housed in facilities differing in their 'dirtiness'. We found evidence for sequential naive to central memory to effector memory development, and confirmed that both memory subsets are heterogeneous in their rates of turnover. We also inferred that early exposure to self and environmental antigens establishes persistent memory populations at levels determined largely, although not exclusively, by the dirtiness of the environment. After the first few weeks of life, however, these populations are continuously supplemented by new memory cells at rates that are independent of environment.

*For correspondence:
ajy2115@cumc.columbia.edu
(AJY);
benedict.seddon@ucl.ac.uk (BS)

†These authors contributed equally to this work
‡These authors also contributed equally to this work

Competing interests: The authors declare that no competing interests exist.

## Introduction

Conventional memory T cells are defined as lymphocytes which respond rapidly to previously encountered epitopes (*Gourley et al., 2004*; *Kaech and Wherry, 2007*). In mice, memory T cells exhibit considerable heterogeneity in their function, circulation patterns, response to re-challenge, and capacities for proliferative self-renewal and survival (*Farber, 2000*; *Kaech and Wherry, 2007*; *Jameson and Masopust, 2009*; *Gossel et al., 2017*). This phenotypic heterogeneity is reflected in differential expression of various cell-surface molecules. In uninfected naive mice, there are at least two distinct populations of recirculating cells distinguished by their expression of the lymph node homing receptor L-selectin (CD62L); CD44[hi] CD62L[−] effector memory (T_EM) and CD44[hi] CD62L[+] central memory (T_CM) cells. During immune responses to active infection, there is an even more complex mix of effector and memory intermediates (*Jameson and Masopust, 2018*).

While it is clear that memory to infection resides amongst these CD44[hi] subsets, it is also evident that they are generated in naive mice in the absence of overt infection. The functional significance of these memory-phenotype (MP) CD4[+] T cells is not fully understood, but there is evidence they can augment primary immune responses. They can facilitate rapid production of IFN-γ during the early inflammatory phase of the immune response to *Toxoplasma gondii* infection and enhance T_H1-type CD4[+] T cell responses later in infection (*Kawabe et al., 2017*). There is also evidence that MP cells are capable of making rapid cross-reactive responses during primary infections (*Min and Paul, 2005*). Given that MP cells represent the majority of the memory compartment in specific pathogen-

free (SPF) mice (*Kawabe et al., 2017*), a better understanding of how these cells are generated and maintained is crucial for better understanding their function and impact upon conventional memory to defined challenges.

The precise nature of the forces driving the generation of MP cells remains unclear. Their development appears to require a TCR-mediated activation event; $Cd28^{-/-}$ mice have greatly reduced numbers of MP cells (*Kotani et al., 2006*), and mice lacking canonical NF-$\kappa$B signalling, an obligate pathway in T cell activation, are completely devoid of such cells (*Webb et al., 2019*). Whether the TCR stimuli derive from self or foreign recognition events is unknown. MP cell generation may reflect a stochastic process in which CD4$^+$ naive T cells occasionally encounter homeostatic stimuli that are above an activation threshold (*Sprent and Surh, 2011*), and indeed MP cells are generated not only in lymphopenia but also constitutively under replete conditions throughout life (*Gossel et al., 2017*; *Kawabe et al., 2017*). There is a positive correlation between the affinity of naive CD4$^+$ T cells to self-antigens and the potential for differentiating into MP cells (*Kawabe et al., 2017*), although self-reactivity is also positively correlated to reactivity to foreign antigens (*Mandl et al., 2013*). Stronger evidence supporting an autoreactive stimulus comes from the failure of broad-spectrum antibiotics to prevent conversion of naive T cells to MP cells following adoptive transfer in vivo, and the observation that mice raised in germ-free conditions contain similar numbers of MP cells as those in SPF conditions (*Kawabe et al., 2017*). However, there is also a role for foreign environmental antigen in generating T cell memory compartments, since mice raised in more antigenically diverse environments, but still in the absence of overt infection, exhibit larger peripheral memory CD8$^+$ T cell pools than SPF mice (*Beura et al., 2016*). When exactly such environmental stimuli impact upon memory compartment development, or how foreign and self reactivity combine to form the memory compartments, is unknown.

The differentiation pathways of these MP cells are also poorly understood. They may derive either directly from naive T cells or through interconversion of other memory phenotypes. Amongst CD8$^+$-lineage cells, there is evidence that some MP subpopulations are generated in the thymus (*Lee et al., 2011*). In the case of CD4$^+$ lineage cells, it has been suggested that MP cells derive from peripheral naive phenotype cells in a thymus-independent fashion (*Kawabe et al., 2017*). While both CD4$^+$ T$_{CM}$ and T$_{EM}$ are produced constitutively in adult mice (*Gossel et al., 2017*), it remains unclear how their differentiation patterns relate to those of classically antigen-stimulated naive T cells.

In this study, we aimed to characterise the development and maintenance of memory CD4$^+$ T cell subsets in adult mice to identify the nature and timing of the signals driving these dynamics in the absence of overt infection. To do this, we quantified the homeostasis and ontogeny of memory CD4$^+$ T cells in identical strains of mice raised in two different animal facilities with distinct caging environments; those housed in individually ventilated cages (IVCs) and fed irradiated water, and those fed untreated tap water and housed in open cages, who might consequently be exposed to a greater variety of environmental antigens. We made use of an established temporal fate-mapping model in both cohorts, which allowed us to track the development of T cells under replete conditions (*Hogan et al., 2015*; *Gossel et al., 2017*). We also used data from germ-free (GF) mice to dissect further the relative contributions of self and environmental antigens to the generation and maintenance of MP cells.

## Results

### Modelling the ontogeny and homeostasis of CD4$^+$ MP T cell subsets

We employed a system described previously (*Hogan et al., 2015*; *Gossel et al., 2017*) to examine the flow of cells into memory subsets (*Figure 1A*). Briefly, CD45.1$^+$ C57Bl6/SJL host mice of various ages were treated with optimised doses of the transplant conditioning drug busulfan to selectively deplete haematopoietic stem cells (HSC). The HSC compartment was then reconstituted by bone marrow transplantation (BMT) with congenically labelled bone marrow from CD45.2$^+$ C57Bl6/J donors. The progeny of donor HSC were then followed as they developed in the thymus and percolated into the peripheral T cell pools, initially replete with host-derived cells. Total numbers of CD4$^+$

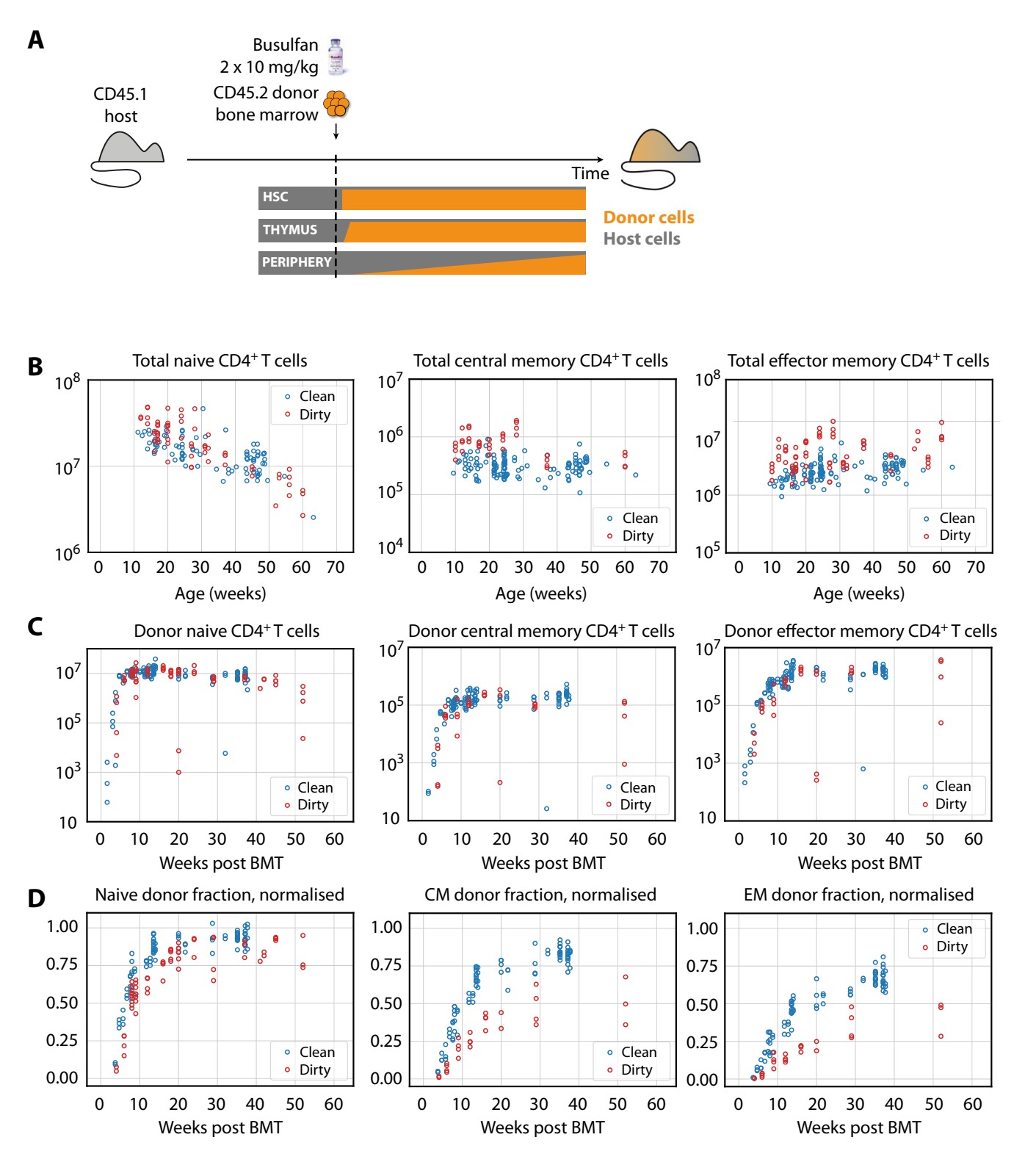

**Figure 1.** Characterising CD4$^+$ T cell subsets in busulfan chimeric mice in clean and dirty environments. (**A**) Generating bone marrow chimeras to map T cell homeostasis. Donor T cells (orange) differentiate and percolate through the thymus and periphery, replacing host cells (grey). (**B**) Comparing total numbers of CD4$^+$ naive, T$_{CM}$ and T$_{EM}$ cells in clean and dirty chimeric mice. (**C**) Numbers of donor-derived CD4$^+$ subsets recovered from spleen and lymph nodes of clean and dirty chimeric mice. Bone marrow transplants (BMT) were performed in mice between ages 5–26 weeks. (**D**) The donor

*Figure 1 continued on next page*

*Figure 1 continued*

fraction (chimerism) within CD4$^+$ T cell subsets, varying with time post BMT, normalised to the chimerism in the double-positive thymocytes in each mouse.

The online version of this article includes the following figure supplement(s) for figure 1:

**Figure supplement 1.** Busulfan chimeric mice exhibit normal numbers of CD4$^+$ naive, central and effector memory T cells.

**Figure supplement 2.** Empirical descriptions of the size and chimerism of the putative source populations for CD4$^+$ T$_{CM}$ and T$_{EM}$ in adult mice.

naive T cells and T$_{CM}$ and T$_{EM}$ cells in these busulfan chimeric mice were normal, in comparison to untreated wild-type (WT) controls (*Figure 1—figure supplement 1*), confirming that the busulfan treatment regime left peripheral compartments intact (*Hogan et al., 2015*; *Gossel et al., 2017*). The kinetics with which donor-derived cells infiltrate the peripheral compartments – first naive, and then memory subsets – are rich in information regarding developmental pathways, rates of turnover and differentiation of lymphocyte populations, and any heterogeneity in homeostatic dynamics within them (*Hogan et al., 2015*; *Gossel et al., 2017*; *Rane et al., 2018*).

We studied busulfan chimeric mice from two housing facilities that employed different levels of mouse containment. At the MRC National Institute for Medical Research (NIMR), mice were held in open cages and fed untreated tap water, while mice held at the UCL Comparative Biology Unit (UCL) were maintained in individually ventilated cages (IVCs) and fed irradiated water. Henceforth, we refer to UCL sourced mice as 'clean' and NIMR sourced mice as 'dirty', in reference to the presumed difference in health status of the mice. We use these terms for clarity, but emphasise that they are relative; mice co-housed with pet-store or feral mice would be expected to be substantially 'dirtier' (*Beura et al., 2016*), and those in turn are cleaner than truly feral mice. In both environments, the same C57Bl6/SJL strain was analysed by the same researcher and cells were enumerated using the same single CASY counter. From age 10 weeks onwards, the numbers of CD4$^+$ naive T cells in mice from clean and dirty environments were broadly similar (*Figure 1B*, left panel). The total sizes (host+donor) of all circulating memory CD4$^+$ T cell subsets remained relatively stable over the time frame of analysis, but were significantly larger in dirty mice (*Figure 1B*, right panels) at age 10 weeks. Following BMT, donor-derived memory T cells accumulated in similar numbers in the two environments (*Figure 1C*). Therefore, these two observations result in a lower proportional replacement of pre-existing memory cells with donor memory cells in dirty mice (*Figure 1D*).

To quantify the cellular processes underlying these kinetics, we first considered a simple mechanistic explanation shown schematically in *Figure 2A*. In this 'homogeneous' model, each memory population (CD4$^+$ T$_{CM}$ or T$_{EM}$) is fed at a constant *per capita* rate from a precursor population (source). We refer to this rate as the force of recruitment, $\varphi$. The total cellular flux into a memory subset per day is then $\varphi$ multiplied by the size of the source population, which in principle could be CD4$^+$ naive T cells, or the complementary memory subset. We assume that memory cells are then lost at a constant net *per capita* rate $\lambda$, which is the balance of loss (turnover) and proliferative self-renewal. In particular, the 'clonal half-life' $\ln(2)/\lambda$ is the average time taken for a population that undergoes any degree of self-renewal to halve in size, and may be much longer than the lifespan of any particular cell within it.

We also considered a 'two phase' model of memory dynamics (*Figure 2B*), which was motivated by three observations. First, newly generated donor CD4$^+$ T$_{CM}$ and T$_{EM}$ in busulfan chimeras express Ki67, a marker of recent cell division, at higher levels than their established host-derived counterparts for some time after BMT (*Gossel et al., 2017*), although these levels eventually converge (data not shown). These observations suggest that memory CD4$^+$ T cell populations become less proliferative, on average, with time since entry into the compartment. Second, we previously found evidence, using BrdU labelling in WT mice, that both CD4$^+$ T$_{EM}$ and T$_{CM}$ appear to be kinetically heterogeneous, comprising at least two subpopulations turning over and dividing at different rates (*Gossel et al., 2017*). Third, and consistent with this picture, the increases in donor chimerism that we observed in both T$_{CM}$ and T$_{EM}$ with time post-BMT were suggestive of a biphasic kinetic, with a relatively rapid accumulation of donor cells followed by a slower increase (*Figure 1D*). As shown in Materials and methods, the rate of accumulation of new memory cells is dictated by both the dynamics of influx and the net loss rate of existing memory, $\lambda$. Therefore, all three observations are consistent with a mechanism in which cells newly recruited into memory comprise a subpopulation

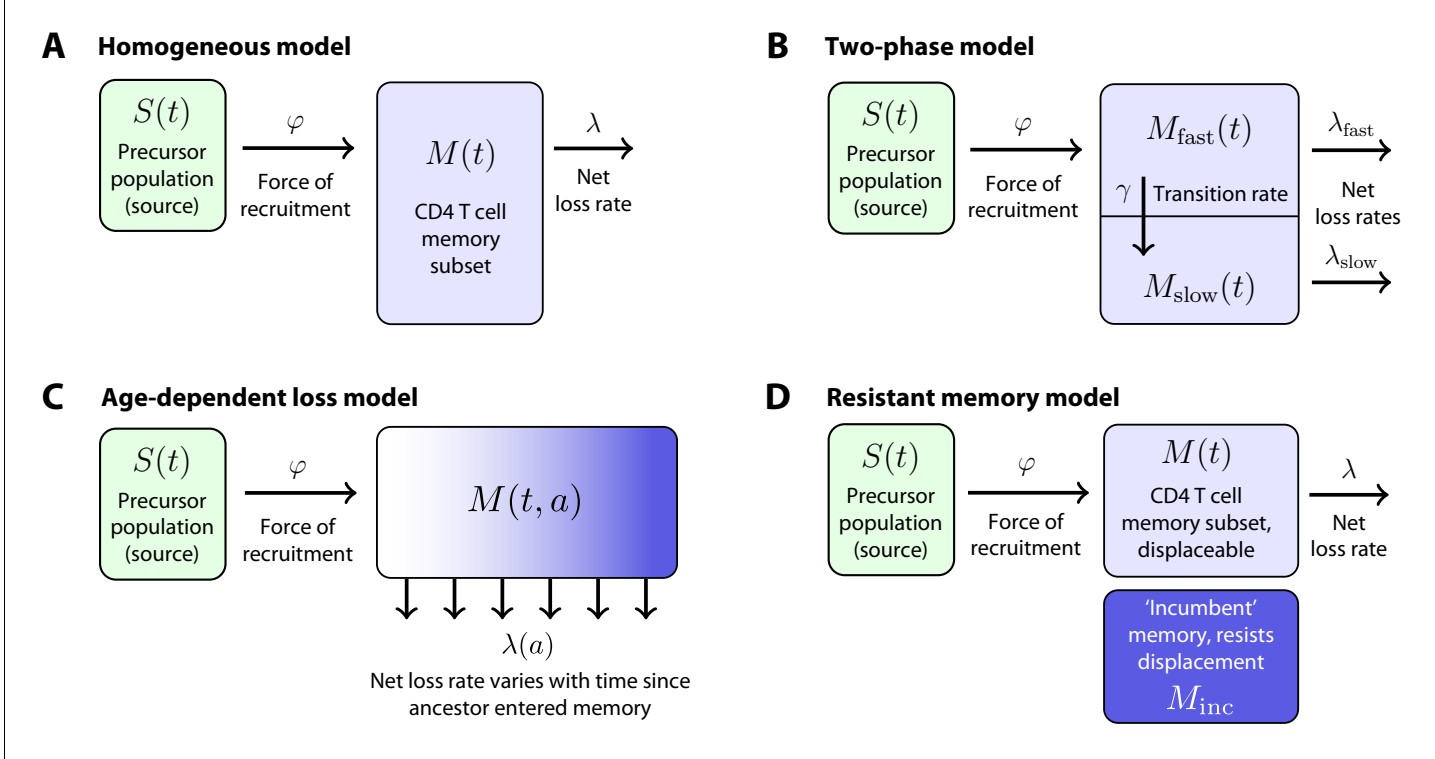

**Figure 2.** Models of the generation and maintenance of memory CD4$^+$ T cell subsets in adult mice. (A) New cells from a precursor (source) population of size $S(t)$ flow in to a homogeneous memory subset $M(t)$ at total rate $\varphi S(t)$. The force of recruitment $\varphi$ is approximately the daily probability that any given cell from the source will be recruited into memory, multiplied by an expansion factor. This memory population may self-renew through division and be lost through death or differentiation and is continually supplemented by cells from the source. We assume that the net loss rate (loss minus division) is a constant, $\lambda$. (B) In a two-phase model of memory, new cells are recruited at rate $\varphi S(t)$ into a population $M_{\text{fast}}(t)$ that has a high net loss rate $\lambda_{\text{fast}}$ and so is replaced by donor cells relatively quickly. These cells transition into a slower subset $M_{\text{slow}}(t)$ at constant rate $\gamma$ and are then lost at net rate $\lambda_{\text{slow}}<\lambda_{\text{fast}}$. (C) The age-dependent loss model; here, the net loss rate of memory is a continuous function of cell 'age' $a$, defined as the time since a cell or its ancestor entered the memory pool. The model tracks the evolution of the population density of memory T cells of age $a$ at host age $t$, $M(t,a)$. (D) The resistant memory model invokes a subpopulation of 'incumbent' memory cells which are presumed to be established early in life, stable in numbers, and not replenished from the source population. As in the homogeneous model, the source feeds a compartment of 'displaceable' cells, with net loss rate $\lambda$.

that both divides rapidly and has a high net loss rate $\lambda_{\text{fast}}$. These cells then transition to a more quiescent state that divides more slowly and also has a lower net loss rate, $\lambda_{\text{slow}}<\lambda_{\text{fast}}$ (*Figure 2B*). The transient differences in Ki67 expression in donor and host memory cells could then be explained by an enrichment for 'new' memory (donor cells) in the fast phase in the weeks immediately following BMT, and not by any intrinsic differences in the behaviour of donor and host cells. Further, this transient difference in Ki67 expression implies a linear flow from fast to slow, rather than a branched process of establishment of the two populations separately; in the latter case, we would expect no differences in Ki67 expression between host and donor cells at any time, provided Ki67 expression within each subset derives entirely from self-renewal and is not inherited from the precursor population.

While the two-phase model is perhaps a minimal description of these observations, it seems plausible that any transition from active to quiescent memory might be more continuous. We previously found evidence for smooth changes in the rates of division and/or loss of naive T cells with their post-thymic age (*Rane et al., 2018*). We therefore also considered a model in which the net loss rate $\lambda$ of a cohort of cells changes continuously with the time since their common ancestor entered memory, $a$ (the 'age-dependent loss model', *Figure 2C*). While the observations above are most consistent with $\lambda_{\text{slow}}<\lambda_{\text{fast}}$, or a decreasing $\lambda(a)$, when fitting the two-phase and age-dependent loss models we placed no constraints on their parameters and allowed the data to determine their

values. When analysing the age-dependent loss model, we also explored a variety of forms for $\lambda(a)$ (see Materials and methods).

Finally, we considered an alternative form of heterogeneity in memory, in which subpopulations of CD4$^+$ T$_{CM}$ and T$_{EM}$ generated early in life persist and are not replenished by newer cells (*Gossel et al., 2017*; *Figure 2D*). These 'incumbent' or 'resistant memory' populations, assumed to be stable in numbers and entirely host-derived (that is, established before 5 weeks of age, the earliest age at BMT in this study), could naturally explain the limited donor chimerism within memory subsets and, if they are less dynamic than memory generated later in life, might also be able to explain host/donor differences in Ki67 expression.

### The kinetics of accumulation of CD4$^+$ MP cells are consistent with a naive → T$_{CM}$ → T$_{EM}$ pathway, and both memory subsets are heterogeneous in their turnover

We compared the abilities of the four mechanisms to describe the replacement kinetics of memory subsets in the dirty and clean environments. The kinetic of donor chimerism in CD4$^+$ T$_{EM}$ clearly lagged that of CD4$^+$ T$_{CM}$ (*Figure 1D*), ruling out T$_{EM}$ as a direct predictor of T$_{CM}$ accumulation. We therefore considered only naive T cells as the source for T$_{CM}$, but considered both naive and T$_{CM}$ cells as potential sources of T$_{EM}$.

For each combination of source, environment (clean/dirty), and subset (T$_{EM}$/T$_{CM}$), we fitted each model simultaneously to the timecourses of the total cell numbers and the proportion of donor cells within the subset using a maximum likelihood approach. We then calculated the combined probabilities (joint likelihoods) that the replacement kinetics of a given subset in both clean and dirty environments derive from each combination of source and model, allowing for different parameters in clean and dirty mice. We then compared the support for each combination using the Akaike Information Criterion (*Table 1*, values in bold). Details of the model formulation, model fitting and inference procedures are given in Materials and methods.

We found clearly stronger support for T$_{CM}$ cells rather than naive T cells as a predictor of T$_{EM}$ production (*Table 1*). This conclusion contrasts with that of our earlier study (*Gossel et al., 2017*), which found evidence for a direct naive → T$_{EM}$ transition; however, while these inferences may be model-dependent to some extent, the more detailed timecourses we studied here gave us greater power to discriminate between the two pathways.

We found almost no support for the homogeneous or resistant memory models. For T$_{CM}$ the age-dependent loss was strongly favoured statistically, while for T$_{EM}$ the two-phase model had the strongest support (fits shown in *Figure 3*). However, the two models gave visually very similar descriptions of each dataset (not shown). This similarity is perhaps unsurprising, as both describe a progressive increase in clonal persistence the longer cells or their progeny reside within memory. Therefore, we remain somewhat equivocal regarding the true nature of heterogeneity in each, and present parameter estimates below for both models (*Table 2*). A robust conclusion, however, is that we find a progressive lengthening of clonal lifetimes in both memory subsets and environments,

**Table 1.** Measures of support (using differences in the corrected Akaike information criterion, AICc - AICc$_{min}$; see Materials and methods) for models in which CD4$^+$ T$_{CM}$ derive directly from CD4$^+$ naive T cells, and T$_{EM}$ derive either from naive T cells or T$_{CM}$. AICc differences are shown in bold, with zero indicating the model with strongest support, and positive differences representing reduced support, with differences of 10 or more generally considered highly significant. Figures in parentheses are the log likelihoods, reflecting the quality of fit of each model. We also show the number of parameters estimated for each model in each environment.

| | | CD4$^+$ central memory | CD4$^+$ effector memory | |
| | | Source population | Source population | |
| Model | Parameters | CD4$^+$ naive | CD4$^+$ naive | CD4$^+$ central memory |
|---|---|---|---|---|
| Homogeneous | 3 | **128** (10) | **89** (75) | **160** (39) |
| Two-phase | 5 | **9** (74) | **29** (109) | **0** (124) |
| Age-dependent loss | 4 | **0** (76) | **29** (107) | **10** (117) |
| Resistant memory | 4 | **26** (63) | **45** (99) | **49** (97) |

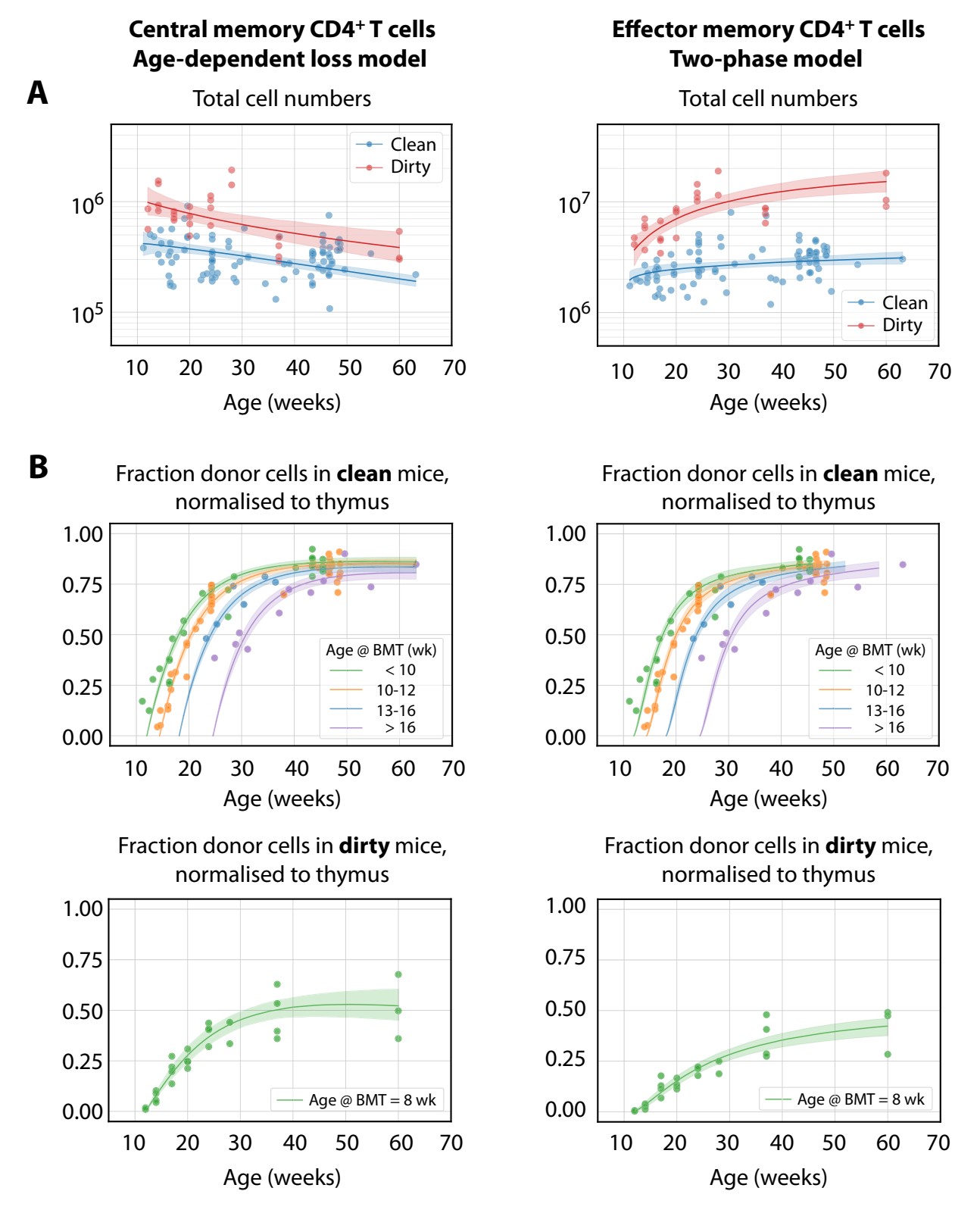

**Figure 3.** The best-fitting models of CD4+ MP T cell dynamics. The age-dependent loss model was the best description of CD4+ $T_{CM}$ dynamics, and the two-phase model best described $T_{EM}$. (**A**) Total (donor+host) numbers of memory T cells and (**B**) chimerism, from 4 weeks post bone marrow transplant (BMT). To visualise the fits, clean facility mice were grouped into small ranges of age at BMT, and the four curves show the model predictions for the median age within each group. All trajectories are described with the same parameters, differing only in the kinetics of the source population,

*Figure 3 continued on next page*

*Figure 3 continued*

which is age-dependent. The lower panels show the fitted trajectories of CD4$^+$ T$_{CM}$ and T$_{EM}$ chimerism in mice in the dirty environment, all of which underwent BMT at a similar age.

with newly recruited memory being lost on timescales of days to weeks, and more established memory persisting for several months (***Figure 4***).

## Constitutive generation of circulating CD4$^+$ T$_{EM}$ and T$_{CM}$ cells in adult mice occurs at constant rates, irrespective of environment

The kinetics of accumulation of donor-derived memory cells were visually indistinguishable in the two environments from age 10 weeks onwards (***Figure 1C***), suggesting similar net rates of recruitment and loss. Consistent with this observation, we found no significant differences between clean and dirty mice in the forces of recruitment ($\varphi$) into circulating CD4$^+$ memory T cell subsets between the ages of 11–64 weeks, in either model, and no substantial differences in their rates of loss (***Figure 4*** and ***Table 2***). Therefore, we infer that antigenic stimuli common to both environments drive the continuous generation of new CD4$^+$ effector and central memory T cells in adult mice, at total rates that are proportional to the sizes of their precursor populations.

## Quantifying the long-term dynamics of CD4$^+$ MP T cell subsets

Our analyses give a quantitative picture of recruitment into memory and the cells' subsequent life-histories, and allow us to identify features of their population dynamics that are common to both environments and model-independent. First, the donor chimerism in T$_{CM}$ reached substantially lower levels than that of their CD4$^+$ naive T cell precursors (***Figure 1D***), suggesting that the rate of generation of new memory in both environments wanes with age, and/or that more established memory

**Table 2.** Estimates of parameters governing CD4$^+$ T$_{CM}$ and T$_{EM}$ homeostasis in adult mice.

| Model | Parameter | Quantity | CD4$^+$ T$_{CM}$ | | CD4$^+$ T$_{EM}$ | |
| --- | --- | --- | --- | --- | --- | --- |
| | | | Clean | Dirty | Clean | Dirty |
| Two-phase | Force of recruitment (d$^{-1}$) | $\varphi$ | $1.4\ (1.2, 2.3) \times 10^{-3}$ | $1.5\ (0.77, 54) \times 10^{-3}$ | $1.2\ (0.76, 12)$ | $1.1\ (0.29, 23)$ |
| | Daily cell influx at age $t^* = 20$ wk | $\varphi S(t^*)$ | $2.3\ (2.0, 3.9) \times 10^{-3}$ | $3.2\ (1.7, 120) \times 10^4$ | $36\ (22, 370) \times 10^4$ | $84\ (21, 1700) \times 10^4$ |
| | Net loss rate of fast subset (d$^{-1}$) | $\lambda_{\text{fast}}$ | $0.082\ (0.063, 0.14)$ | $0.10\ (0.043, 5.2)$ | $0.23\ (0.12, 2.4)$ | $0.39\ (0.045, 9.0)$ |
| | Net loss rate of slow subset (d$^{-1}$) | $\lambda_{\text{slow}}$ | $5.9\ (3.0, 9.4) \times 10^{-3}$ | $4.8\ (2.7, 6.9) \times 10^{-3}$ | $2.5\ (0.81, 4.6) \times 10^{-3}$ | $4.8\ (1.0, 8.8) \times 10^{-3}$ |
| | Clonal half-life of fast subset (d) | $\ln(2)/\lambda_{\text{fast}}$ | $8.4\ (5.0, 11)$ | $6.7\ (0.14, 16)$ | $3.0\ (0.30, 5.8)$ | $1.8\ (0.078, 16)$ |
| | Clonal half-life of slow subset (d) | $\ln(2)/\lambda_{\text{slow}}$ | $120\ (75, 230)$ | $140\ (101, 260)$ | $270\ (150, 770)$ | $140\ (75, 570)$ |
| | % of memory transitioning to slow | $100\gamma/(\lambda_{\text{fast}} + \gamma)$ | $3.4\ (0.46, 4.1)$ | $5.0\ (0.18, 10)$ | $1.8\ (0.21, 2.4)$ | $9.1\ (0.36, 38)$ |
| | Proportion slow at $t^* = 20$ wk | $M_{\text{slow}}(t^*)/M(t^*)$ | $0.25\ (0.12, 0.30)$ | $0.61\ (0.45, 0.75)$ | $0.36\ (0.25, 0.41)$ | $0.72\ (0.41, 0.87)$ |
| Age-dependent loss | Force of recruitment (d$^{-1}$) | $\varphi$ | $0.43\ (0.38, 0.57) \times 10^{-3}$ | $0.26\ (0.18, 0.43) \times 10^{-3}$ | $0.10\ (0.086, 0.13)$ | $0.062\ (0.049, 0.26)$ |
| | Daily cell influx at age $t^* = 20$ wk | $\varphi S(t^*)$ | $0.70\ (0.62, 0.93) \times 10^4$ | $0.57\ (0.39, 0.96) \times 10^4$ | $3.0\ (2.5, 3.8) \times 10^4$ | $4.6\ (3.7, 19) \times 10^4$ |
| | Net loss rate of new memory (d$^{-1}$) | $\lambda_0$ | $2.2\ (2.0, 2.7) \times 10^{-2}$ | $1.2\ (0.79, 2.0) \times 10^{-2}$ | $1.2\ (0.91, 1.5) \times 10^{-2}$ | $1.1\ (-44, 26) \times 10^{-5}$ |
| | Memory age threshold* (d) | $A$ | $150\ (140, 200)$ | $190\ (130, 310)$ | $150\ (130, 230)$ | NA |

95% confidence intervals are shown in parentheses. *In the age-dependent loss model, the threshold cell age $A$ defines the beginning of the more persistent phase of memory maintenance ($\lambda(a) = \lambda_0/(1 + (a/A)^3)$); for T$_{EM}$ in dirty mice, estimates of $\lambda_0$ were close to zero, and $A$ was poorly constrained.

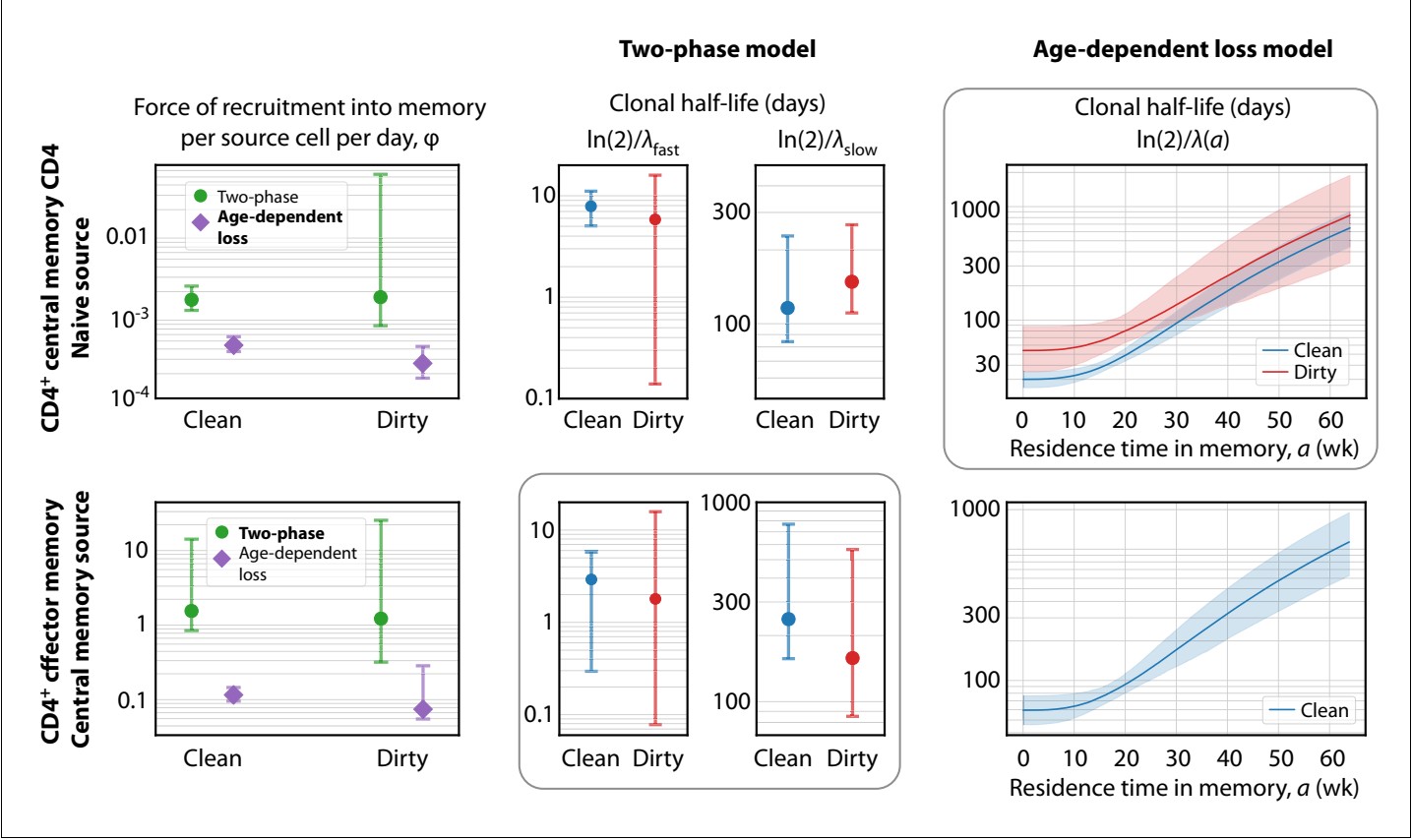

**Figure 4.** Key parameters describing the constitutive production of central and effector memory CD4+ T cells in adult mice. Left panels: Estimates of the force of recruitment from the source ($\varphi$) for each model and each population. Vertical bars represent bootstrapped 95% confidence intervals. The favoured model for each population is indicated in bold in the legend. Middle panels: The estimated clonal half-lives of fast and slow memory in the two-phase model. The enclosing box indicates that this model was favoured for $T_{EM}$. Right panels: Estimates of the clonal half-lives, which vary with cell age, derived from the age-dependent loss model, favoured for $T_{CM}$. For $T_{EM}$ in dirty mice, the estimated $\lambda(a)$ was close to zero and the clonal half-life is not shown. Shaded bands indicate the range of predicted half-lives arising from the 95% confidence intervals on $\lambda(a)$. All parameter estimates are given in *Table 2*.

The online version of this article includes the following figure supplement(s) for figure 4:

**Figure supplement 1.** Predicted survival curves for populations entering CD4+ MP T cell compartments using the age-dependent loss model.

has a competitive advantage over recently recruited cells. We find evidence for both processes here. We show in Materials and methods that if influx declines faster than the average rate of turnover, a population will be unable to reach the same level of chimerism as its precursor – in effect, the flow from the source 'dries up' more quickly than the memory cells can be replaced by immigrants. We see signs of this effect; due to thymic involution, CD4+ naive T cell numbers decay exponentially in both clean and dirty adult mice (*Figure 1B*) with population half lives of 228 days (95% CI 227–231 days) and 143 (142-144) days, respectively. These timescales are comparable to or shorter than the estimated half lives of established $T_{CM}$ memory clones ($\ln(2)/\lambda_{slow}$ in the two-phase model; and $\ln(2)/\lambda(a)$ for $a > 30$ weeks in the age-dependent loss model; *Figure 4* and *Table 2*). In addition, both the two-phase and age-dependent models indicate that older memory clones have a fitness advantage over newer ones. Therefore, the limited replacement of host CD4+ $T_{CM}$ by donor cells derives from the decline in naive T cell numbers with age, slow average rates of turnover, and the increased persistence of more established memory cells. This slow rate of accumulation of new $T_{CM}$ in turn acts to limit the chimerism observed in $T_{EM}$, which are also lost slowly.

One can also quantify the fates of populations after entering memory, although here our insights are more model-dependent. The two-phase model predicts that the establishment of memory is relatively inefficient, with 'fast' populations lost over timescales of days and only a small proportion of

these surviving to become more persistent 'slow' memory (~2–10% of $T_{EM}$, and ~3–5% of $T_{CM}$; *Table 2*). Despite this inefficiency, the substantial constitutive influxes maintain the fast and slow populations at comparable sizes, consistent with our previous analysis of BrdU labelling of CD4+ $T_{CM}$ and $T_{EM}$ in WT mice (*Gossel et al., 2017*). In contrast, the age-dependent loss model makes lower estimates of the force of recruitment into memory (*Figure 4*, left panels) but predicts more efficient establishment, with newly generated memory having clonal half-lives of 20–40 days and a much greater proportion persisting longer term (*Figure 4—figure supplement 1*).

## Larger memory populations in dirty mice derive from early antigen exposure

Given the similarity of the rates of generation of memory in clean and dirty adult mice, and of their rates of turnover, we infer that the larger, relatively stable $T_{CM}$ and $T_{EM}$ populations in dirty mice (*Figure 3A*) must derive from their establishment in greater numbers in the first few weeks of life. The differences in compartment sizes in the two environments are then sustained well into adulthood by the very slow loss of these early memory populations.

To explore this hypothesis, we used the parameters estimated in adult mice to predict the development of their CD4+ MP T cell populations early in life. To do this, we drew on measurements of naive and central memory CD4+ T cell numbers recovered from the spleen and lymph nodes of WT mice aged between 5 days and 14 weeks, kept in the clean facility (*Figure 5A*). We then used the naive T cell timecourse with the parameters estimated for the best-fitting (age-dependent loss) model of $T_{CM}$ development in clean adult mice, to predict their accumulation of $T_{CM}$ from age 5d onwards (*Figure 5B*, left panel), starting from the mean numbers of CD4+ $T_{CM}$ observed at age 5d. The model slightly underestimated $T_{CM}$ numbers in clean adult mice and failed to capture their rapid accumulation up to age 4 weeks. In contrast, using the observed timecourse of $T_{CM}$ in clean WT mice as a source (*Figure 5A*, right panel), the establishment of the $T_{EM}$ compartment was predicted remarkably well by the adult parameters from the favoured two-phase model (*Figure 5B*, right panel).

To predict the early kinetics of CD4+ MP T cell populations in dirty mice, we first assumed their accumulation of naive T cells closely approximated that in clean mice, given that naive T cell numbers were similar in young adults from the two facilities (*Figure 1B*). We then used this timecourse (*Figure 5A*, left panel) with the parameters from the best-fitting (age-dependent loss) model of $T_{CM}$ development in adult dirty mice, to predict their kinetics of accumulation. This prediction underestimated $T_{CM}$ numbers at age 14 weeks by a factor of approximately 4 (*Figure 5C*, left panel). In turn, using this trajectory as the source of $T_{EM}$, and using the force of recruitment and loss rates derived from adults, led to a similarly substantial underestimate of their numbers (*Figure 5C*, right panel). Predictions using the alternative models in all cases were even poorer; and all predictions were insensitive to the presumed numbers of $T_{CM}$ or $T_{EM}$ at age 5d, which are small and rapidly outnumbered by the influx of new memory cells from their precursor population.

We conclude that to account for memory T cell numbers in adulthood, mice in the clean facility experience a slightly elevated force of recruitment into $T_{CM}$ early in life; and this force is much larger in dirtier mice, presumably deriving from greater levels of exposure to environmental antigens.

## Analysis of germ free mice confirms roles for both autoreactive and commensal stimuli in the establishment of memory compartments

We found that the rate of constitutive recruitment into memory in adult mice was insensitive to variations in environmental commensals, and that these antigens must exert their biggest influence on the establishment of MP cells in neonates and young mice. However, earlier studies reported that numbers of memory cells in the spleens of SPF and germ-free (GF) mice are similar and argue that self-recognition is therefore the sole driver of MP cell generation in early life (*Kawabe et al., 2017*). To reconcile these apparent differences, and dissect the contributions of self and foreign antigens to the establishment of MP T cells in young mice, we compared the size and behaviour of memory CD4+ T cell subsets in C57Bl6/J and/or C57Bl6/SJL mice housed in a wider range of environments. In addition to the clean (UCL) and dirty (NIMR) mice analysed above, we enumerated cells from GF and SPF mice obtained from the Kennedy Institute (KI) in Oxford. Consistent with these earlier studies, substantial numbers of both $T_{CM}$ and $T_{EM}$ MP cells were recovered from GF mice aged between

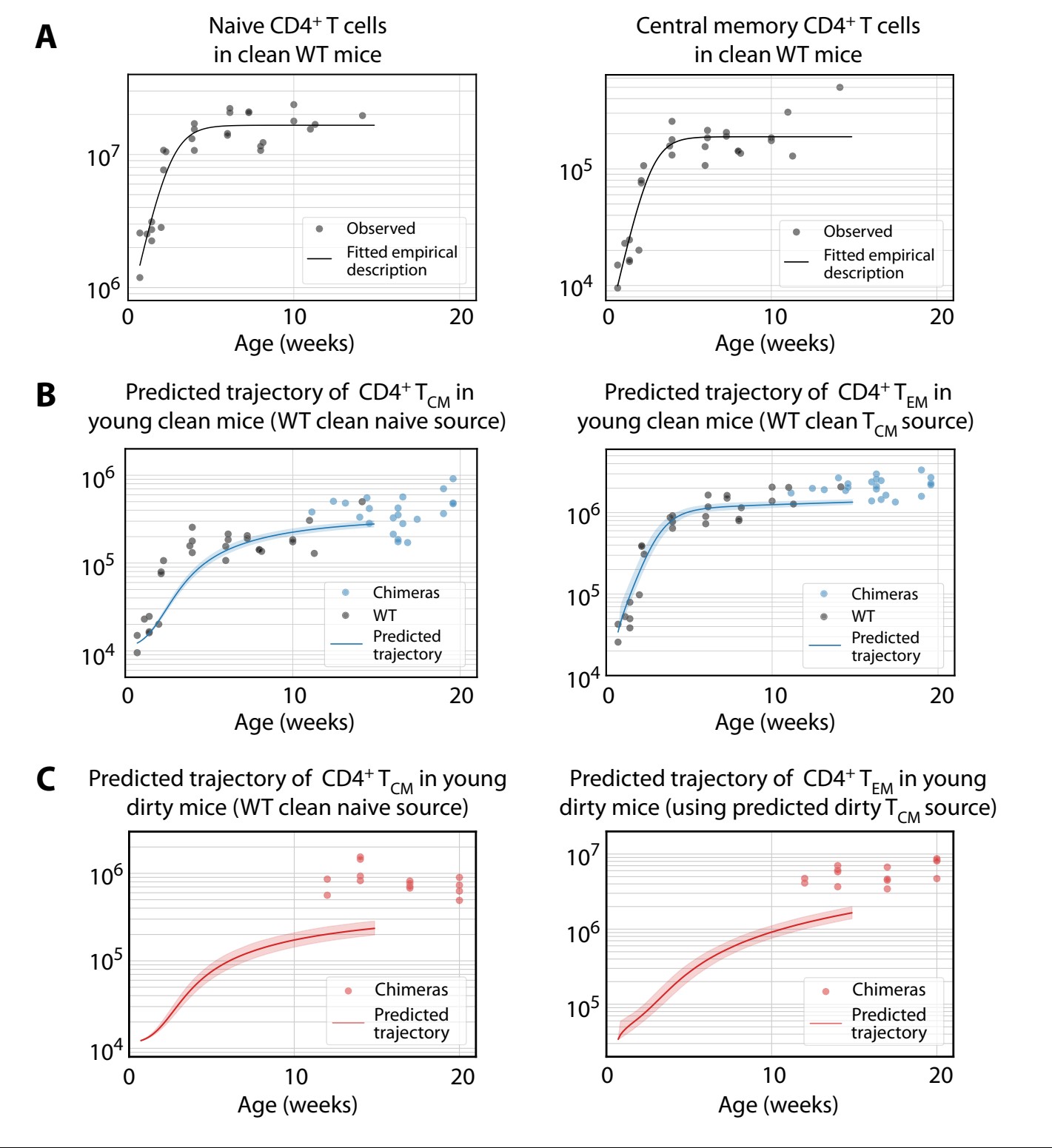

**Figure 5.** Predicting the establishment of CD4+ $T_{CM}$ and $T_{EM}$ in clean and dirty mice. (**A**) The timecourses of numbers of CD4+ naive T cells and $T_{CM}$ recovered from the spleen and lymph nodes of wild-type (WT) mice housed in the clean facility at UCL, aged 5 days to 15 weeks. We fitted a descriptor function $S(t) = S_{max}/(1 + e^{-rt}(S_{max} - S_{min})/S_{min})$ to both, using least squares on the log-transformed observations. (**B**) Using these curves to predict the development of CD4+ $T_{CM}$ and $T_{EM}$ in clean mice using the best-fitting model parameters from adults. Shaded regions indicate the spread of predictions over the 95% confidence intervals of the parameters. The black points (not used for fitting) are the numbers of CD4+ $T_{CM}$ and $T_{EM}$

*Figure 5 continued on next page*

*Figure 5 continued*

recovered from clean WT mice. (**C**) Using CD4$^+$ naive T cells from young clean mice as a source, we used the age-dependent loss model of T$_{CM}$ dynamics in adult dirty mice to predict their accumulation (left panel). This trajectory in turn was used to predict the accumulation of T$_{EM}$ in the same mice (right panel).

40 and 200d, confirming that the generation of MP cells does not depend exclusively on commensal-derived foreign antigens (*Figure 6A*). However, the memory CD4$^+$ T cell compartments of GF mice, enumerated from spleen and lymph nodes combined, were significantly smaller than in the SPF mice in all facilities. Clean mice from KI and UCL had similar-sized memory compartments, and in turn both were substantially smaller than those in mice from the dirty facility (NIMR) (*Figure 6A*). Indeed, the dirty mice played host to around five times the number of MP cells found in GF mice, indicating that antigens from commensal organisms are a substantial driver of MP cell expansion.

We also analysed the proliferative activity of MP cells in mice from the different environments, by measuring the expression of Ki67. Division of MP cells is dependent on TCR (*Seddon et al., 2003*) and CD28 costimulation signaling (*Kawabe et al., 2017*). A substantial fraction of MP cells in GF mice expressed Ki67 (*Figure 6B*), indicating cell cycle activity within the previous 3–4 days (*Gossel et al., 2017*). This proportion was broadly similar to that in mice from both clean and dirty environments, indicating that the level of proliferation of CD4$^+$ MP T cells in adult mice was relatively insensitive to environmentally derived stimuli.

## Quantifying the forces exerted by commensals on memory generation early in life

Finally, we estimated the rates of memory generation in young mice in the different environments. We began with the favoured age-dependent loss model of T$_{CM}$ dynamics. Using the parameters from clean adult mice (which were similar to those estimated for dirty mice, and for which no estimates were available for GF mice), and the empirical description of CD4$^+$ naive T cell numbers in clean WT mice (*Figure 5A*), we then estimated the fold changes in the force of recruitment $\varphi$ needed during the first 11 weeks of life needed to seed CD4$^+$ T$_{CM}$ at the average numbers observed in the mice aged between 10 and 28 weeks (*Figure 7*, left panel). CD4$^+$ T$_{CM}$ numbers were relatively stable in all facilities during this period. GF mice needed approximately 0.4 times the force of recruitment in clean adult mice, younger clean mice needed a force approximately 1.1 times greater, and dirty mice required a 2.7-fold increase. As before, uncertainty in memory cell numbers at age 5d had very little effect on the predicted levels of memory attained at week 11, or on these estimated correction factors.

We then used these environment-specific, corrected trajectories of T$_{CM}$ development to predict the accumulation of T$_{EM}$ by age 11 weeks, using the favoured two-phase model. Remarkably, after accounting for the different T$_{CM}$ population sizes, the force of recruitment from T$_{CM}$ to T$_{EM}$ estimated in clean adult mice was also sufficient to account for T$_{EM}$ numbers in all three environments (*Figure 7*, right panel).

In summary, this analysis showed that approximately 2- to 3-fold increases or decreases in the force of recruitment into CD4$^+$ T$_{CM}$ observed in clean adult (UCL) mice were sufficient to explain their numbers in dirty mice from NIMR or GF mice. However the subsequent rate of development of T$_{EM}$ from T$_{CM}$ appeared to be independent of both mouse age and environment, and differences in the numbers of T$_{EM}$ could be explained simply by the differences in the size of the T$_{CM}$ precursor population. These results suggest that the rate of generation of CD4$^+$ T$_{CM}$ from naive T cells in young mice reflects both self antigens and the level of exposure to environmental antigens, but that the rate of differentiation from T$_{CM}$ to T$_{EM}$ is largely insensitive to these forces.

## Discussion

In this study, we compared mice housed in facilities with distinct antigenic burdens to investigate the nature, magnitude and timing of the forces that establish and maintain CD4$^+$ MP T cell compartments. We examined (i) tonic recruitment into the T$_{CM}$ and T$_{EM}$ pools in adults, (ii) the kinetic substructure and maintenance of these compartments throughout life, and (iii) their generation/establishment early in life. Our analyses indicate that self recognition contributes to all these

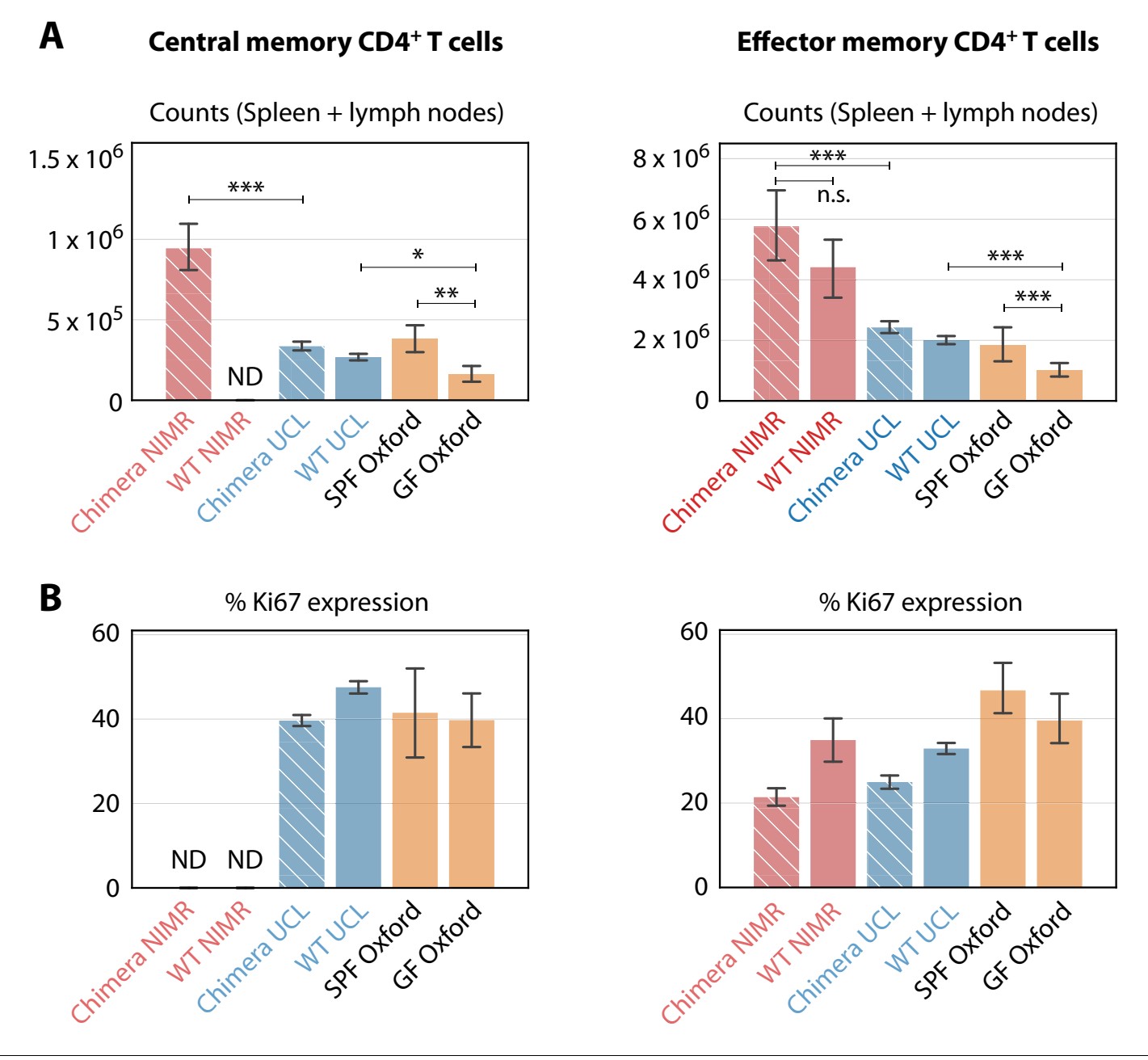

**Figure 6.** Comparing the numbers and proliferative activity (panels A and B respectively) of CD4$^+$ T$_{CM}$ and T$_{EM}$ in adult mice, in different SPF and GF facilities. Cross-hatched bars denote busulfan chimeras, solid bars denote WT mice. NIMR (red bars) and UCL (blue bars) are the 'dirty' and 'clean' facilities used for the bulk of the analysis presented here. 'ND' denotes data not available. * p < 0.05, ** p < 0.01, *** p < 0.001, using the Mann-Whitney test. Group sizes: Panels A; (28, 0, 78, 98, 4, 14) and (46, 11, 78, 98, 4, 14). Panels B; (0, 0, 74, 140, 4, 14) and (18, 4, 74, 140, 4, 14).

processes, but that the contribution of reactivity to commensal antigens is largely restricted to the neonatal period.

Our analysis of GF mice, which lack exposure to commensal organisms, confirmed earlier work showing that generation of MP cells in adults is not driven exclusively by foreign commensals (*Kawabe et al., 2017*) and suggesting that self-recognition is instead the key driver. GF mice are not entirely free of environmental antigens and it is possible that proteins in bedding material and diet could represent foreign antigenic stimuli. However, the argument for self-recognition is made

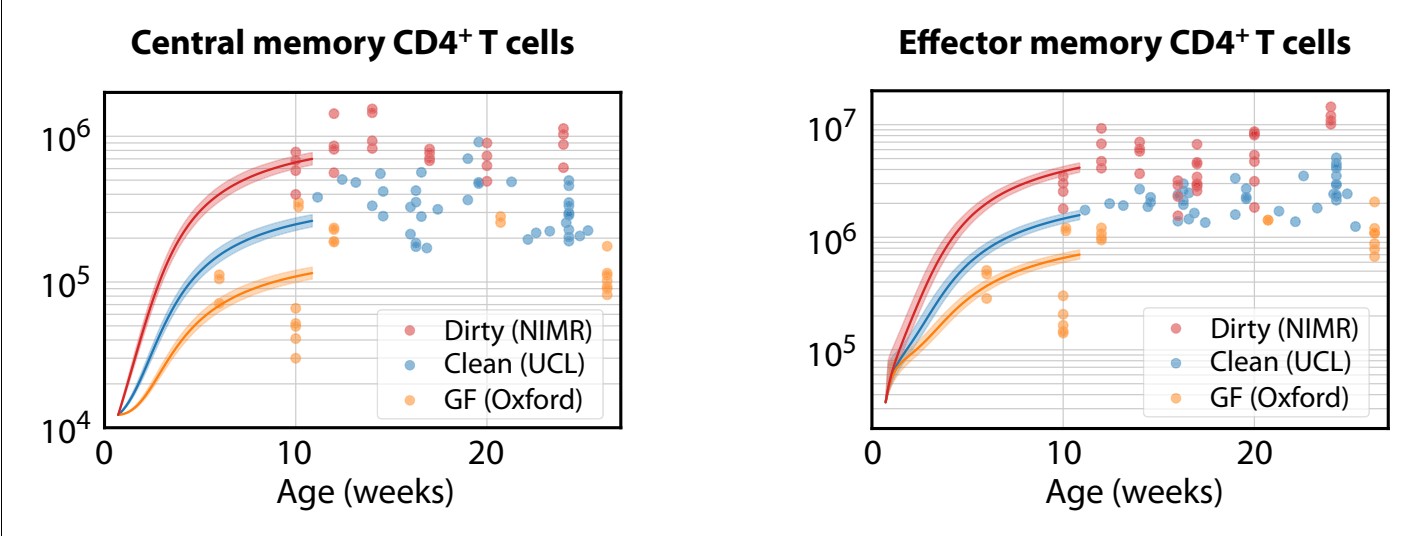

**Figure 7.** Modelling the ontogeny of CD4[+] memory T cell subsets in different facilities. Using the timecourse of CD4[+] naive T cell numbers in young clean WT control mice (**Figure 5A**), and the parameters estimated in clean adult mice, we estimated the corrections to the force of recruitment $\varphi$ needed from birth to age 11 weeks to generate the mean numbers of $T_{CM}$ in adults aged 10–28 weeks in each environment (left panel). These corrected $T_{CM}$ trajectories, together with the force of recruitment and loss rates estimated from clean adult mice, predicted the accumulation of CD4[+] $T_{EM}$ up to age 11 weeks in all three environments (right panel).

through the correlation of the degree of MP cell conversion and steady state proliferation with affinity for self-MHC, as indicated by CD5 expression. We also found that memory CD4[+] T cell division, as reported by Ki67, was substantial and largely independent of the level of commensals. We could not determine the extents to which Ki67 expression derived from the homeostatic proliferation of existing cells or the influx of newly generated (and recently divided) MP cells. However, the common levels of Ki67 across environments, together with our findings from the adult clean and dirty chimeric mice that the rates of memory generation and net loss were insensitive to environment, support our conclusion that self-recognition is the major driver of both recruitment and proliferative renewal in adult memory compartments.

Commensals did, however, have a substantial impact upon the sizes of the memory compartments generated early in life. CD4[+] $T_{CM}$ and $T_{EM}$ numbers in mice raised in dirty environments were 3–5 times greater than those in either GF or cleaner IVC or SPF facilities, and these differences could be explained by differences in the forces of recruitment of CD4[+] $T_{CM}$ during ontogeny. The estimated seven-fold difference in this force between GF and dirty mice prompts the simple interpretation that self-recognition only accounts for ~14% of the memory compartment in dirty mice.

We ascribe differences in memory compartment sizes to different commensal burdens, but it is possible that their smaller sizes in GF mice also derive from their smaller lymph nodes. Bacterial stimulation of DCs is required for their migration into lymph nodes, and these DCs are required for their normal development (**Wendland et al., 2011**; **Moussion and Girard, 2011**; **Zhang et al., 2016**). Therefore, it is not straightforward to separate the indirect influence of commensals on lymphoid development from any direct influences upon memory generation. It is also possible that some of the additional force of recruitment in neonates derives not from commensals but from lymphopenia, which can drive naive T cells to acquire a memory phenotype (**Min et al., 2003**). However, this process was demonstrated by transferring naive cells from adults into very young mice; it is possible that such naive cells do not represent the activity of neonatal naive T cells, which are almost exclusively recent thymic emigrants.

We observed that environment not only impacted memory but also naive T cells. Their numbers naturally decline with age, but this decline was almost twice as fast in dirty mice than in genetically identical mice housed in cleaner facilities (**Figure 1—figure supplement 2A**; numbers halve every 228d (95% CI 227–231) in clean mice, and 143d (142-144) in dirty mice). This difference likely derives from a more rapid reduction in thymic output with age in the dirty environment, and not differences

in lifespans of CD4$^+$ naive T cells in the two environments, because we see similar rates of decline in the numbers of single positive thymocytes at the latest stage of thymic development (halving every every 282d (281-308) in clean mice, 151d (150-159) in dirty mice; data not shown). Therefore, it appears that another consequence of life in a more antigenically diverse environment is more rapid involution of the thymus. It is possible that this effect derives from the stress or inflammation associated with an increased commensal burden, or is somehow a consequence of expanded memory populations; but whatever the mechanism, our data clearly indicate that environmental factors can impact the maintenance of naive T cells.

Our models do not incorporate any homeostatic regulation in the sense of modulation of rates of division or loss through quorum sensing. Since memory cell numbers only vary by a factor of 2–3 between clean and dirty mice, we did not expect to detect any strong variation in net loss rate with pool size with these data, and indeed our estimates of loss rates were similar in the two environments. While we cannot rule it out at higher cell densities, there is arguably little evidence for homeostatic regulation of circulating memory T cells in SPF mice. We observed a range of relatively stable memory compartment sizes in the different containment facilities; these memory compartments do not appear to fill up rapidly with large clones in very young mice, which might occur if division or loss rates are sensitive to total pool size; and they appear to be expandable following multiple infections in older animals (*Vezys et al., 2009*).

The models yielded robust conclusions regarding the nature and magnitudes of the forces generating CD4$^+$ T$_{CM}$ and T$_{EM}$, and the existence of heterogeneity within both subsets, consistent with previous reports by ourselves and others of subpopulations of CD4$^+$ MP cells with distinct rates of division and turnover (*Younes et al., 2011*; *Gossel et al., 2017*). However, these models are abstractions, and resolving the details of kinetic substructure in lymphocyte populations is challenging (*Ganusov et al., 2010*). For CD4$^+$ T$_{CM}$ in adult mice we found evidence for slow and continuous changes in their net loss rates as cells age, and additional support for such a process comes from the slow rate at which Ki67 levels in donor and host memory cells converge in adult busulfan chimeras. In contrast, the data for T$_{EM}$ in adult mice, and the predictions of their accumulation in younger mice, more strongly supported a discrete two-phase model with a relatively rapid transition from fast to slow memory. Our previous study of CD4$^+$ T$_{CM}$ and T$_{EM}$ homeostasis in adult mice used short-term BrdU labelling to identify populations in both subsets that divide and die rapidly (*Gossel et al., 2017*), although in that study we assumed the proliferative and more quiescent pools were maintained independently and so it is not straightforward to compare the rate estimates with those presented here. Overall, although, it seems likely that both MP cell subsets are more heterogeneous than any one of our models suggests. Indeed, there are other potential sources of heterogeneity. One possibility is that MP T cells are generated with a distribution of net loss rates, and those clones with greater intrinsic fitness (lower net loss rates) are simply selected for over time. Such a mechanism – a generalisation of the simple 'resistant memory' model we rejected here – could explain the under-representation of donor cells in the memory compartments of busulfan chimeric mice, and may be difficult to distinguish from our model of gradual changes in fitness with cell age; indeed the two mechanisms are not mutually exclusive. Characterising the homeostatic dynamics of CD4$^+$ memory T cells, and ultimately how these dynamics relate to their functional capacity, requires further study.

## Materials and methods

### Generating busulfan chimeric mice

Mice were treated with optimised low doses of busulfan to deplete HSC but leave peripheral T cell subsets intact. HSC were reconstituted with congenically labelled, T-cell depleted bone marrow to generate stable chimeras (*Figure 1A*). Details of the protocols are given in *Hogan et al. (2017b)* and *Hogan et al. (2017a)*.

### Mice

Busulfan chimeric mice and wild-type control mice were housed in conventional animal facilities, either at the National Institute for Medical Research, London, UK (NIMR); or at the UCL Royal Free Campus, London, UK (UCL). At NIMR, mice were housed in open cages and drank tap water. At

UCL, mice were housed in individually ventilated cages and drank irradiated water. Germ Free and SPF mice were housed at the Oxford Centre for Microbiome Studies, Oxford, UK.

## Flow cytometry

Single cell suspensions were prepared from the thymus, spleen and lymph nodes of busulfan chimeric mice, wildtype control mice, or germ free mice. Cells were stained with the following monoclonal antibodies and cell dyes: CD45.1 FITC, CD45.2 FITC, CD45.2 AlexaFluor700, TCR-$\beta$ APC, CD4$^+$ PerCP-eFluor710, CD44 APC-eFluor780, CD25 PE, CD25 eFluor450, CD25 PE-Cy7, CD62L eFluor450, NK1.1 PE-Cy7 (all eBioscience), CD45.1 BV650, CD45.2 PE-Dazzle, TCR-$\beta$ PerCP-Cy5.5 CD4$^+$ BV711, CD44 BV785, CD25 BV650 (all Biolegend), CD62L BUV737 (BD Biosciences), LIVE/DEAD nearIR and LIVE/DEAD blue viability dyes. For Ki67 staining, cells were fixed using the eBioscience Foxp3/Transcription Factor Staining Buffer Set and stained with either anti-mouse Ki67 FITC or PE (both eBioscience). Cells were acquired on a BD LSR-II or a BD LSR-Fortessa flow cytometer and analysed with Flowjo software (Treestar). Conventional CD4$^+$ cells were identified as live TCR-$\beta$+ CD4$^+$ CD25- NK1.1-, and then CD44 and CD62L were used to identify EM (CD44+CD62L-) and CM (CD44+CD62L+) subsets.

## Modelling the fluxes between naive, central memory and effector memory CD4$^+$ subsets

### The homogeneous model

Our simplest description of the kinetics of the generation and renewal of CD4$^+$ T$_{CM}$ and T$_{EM}$ is illustrated in *Figure 2A* and was formulated as follows. We assume that cells flow into a memory subset of total size $M(t)$ from a precursor population $S(t)$ at total rate $\varphi S(t)$, where $t$ is the age of the animal. The rate constant $\varphi$ is the 'force of recruitment', a compound parameter which is the product of the *per capita* rate of recruitment of cells from the source population per day multiplied by number representing any net expansion that occurs during recruitment. Memory is also lost at net *per capita* rate $\lambda$. This rate is the balance of loss through death and/or differentiation, and any compensatory cell production through division. It represents the rate of decline or growth of a population that self-renews to any extent, rather than the loss rate of individual cells. We place no constraints on this rate, and so $\lambda$ may be positive or negative.

$$\frac{dM(t)}{dt} = \varphi S(t) - \lambda M(t). \tag{1}$$

We assume host and donor cells each obey the same kinetics, so that

$$\frac{dM_{\text{host}}(t)}{dt} = \varphi S_{\text{host}}(t) - \lambda M_{\text{host}}(t) \tag{2}$$

$$\frac{dM_{\text{donor}}(t)}{dt} = \varphi S_{\text{donor}}(t) - \lambda M_{\text{donor}}(t), \tag{3}$$

where the total population size is $M(t) = M_{\text{donor}}(t) + M_{\text{host}}(t)$. Our strategy for parameter estimation was to fit this model simultaneously to the timecourses of total numbers of memory cells $M(t)$, and the donor chimerism within memory, $\chi_M(t)$, which is the fraction of cells in memory that are donor-derived. For reasons detailed below, we normalise this fraction to the proportion of early double-positive (DP1) thymocytes that are donor-derived, which is measured in the same mouse and denoted $\chi_{\text{DP1}}$;

$$\chi_{\text{DP1}} = \frac{\text{DP1}_{\text{donor}}}{\text{DP1}_{\text{host+donor}}}, \quad \chi_M(t) = \frac{M_{\text{donor}}(t)}{M(t)}, \quad \chi_{\text{M,norm}}(t) = \frac{\chi_M(t)}{\chi_{\text{DP1}}}. \tag{4}$$

*Equations (2) and (3)* then give

$$\frac{d}{dt}\chi_{\text{M,norm}}(t) = \frac{d}{dt}\left(\frac{M_{\text{donor}}(t)}{\chi_{\text{DP1}}(t)M(t)}\right) = \frac{1}{\chi_{\text{DP1}}(t)}\frac{d}{dt}\left(\frac{M_{\text{donor}}(t)}{M(t)}\right) - \frac{d\chi_{\text{DP1}}/dt}{\chi_{\text{DP1}}(t)^2}\left(\frac{M_{\text{donor}}(t)}{M(t)}\right). \tag{5}$$

If observations are made sufficiently long after BMT (more than 3–4 weeks), chimerism among DP

thymocytes can be assumed to have stabilised (*Hogan et al., 2015*) and so we can neglect the term in $d\chi_{\text{DP1}}/dt$. Then

$$
\begin{aligned}
\frac{d}{dt}\chi_{\text{M,norm}}(t) &= \frac{1}{\chi_{\text{DP1}}M(t)}\left(\varphi S_{\text{donor}}(t) - \lambda M_{\text{donor}}(t) - \frac{M_{\text{donor}}(t)}{M(t)}\frac{dM(t)}{dt}\right) \\
&= \frac{1}{\chi_{\text{DP1}}M(t)}\left(\varphi S_{\text{donor}}(t) - \lambda M_{\text{donor}}(t) - \chi_M(t)(\varphi S(t) - \lambda M(t))\right) \\
&= \frac{1}{\chi_{\text{DP1}}M(t)}\left(\varphi S_{\text{donor}}(t) - \chi_M(t)\varphi S(t)\right) \\
&= \frac{\varphi S(t)}{M(t)}\left(\chi_{\text{source,norm}}(t) - \chi_{\text{M,norm}}(t)\right),
\end{aligned}
\tag{6}
$$

where we define

$$
\chi_{\text{source,norm}}(t) = \frac{1}{\chi_{\text{DP1}}}\frac{S_{\text{donor}}(t)}{S(t)}.
\tag{7}
$$

By normalising the chimerism of both the source and the memory populations to that in the thymus, we remove any variation in these quantities due to variation across individuals in the degree of chimerism achieved with busulfan treatment and BMT.

Fitting required initial conditions for the total numbers of memory cells and donor chimerism. We solved all of the models from host age $t_0$, which was chosen to be the time at which donor chimerism in memory for the mouse with the youngest age at BMT could be assumed to be zero and donor chimerism in DP1 had stabilised. This was host age 66d for the clean mice and 84d for dirty mice (26d and 28d post-BMT, respectively). Our results were insensitive to changes of a few days in these baseline ages. We also required functional forms for the kinetics of the immediate precursor (source) population $S(t)$ and $\chi_{\text{source,norm}}(t)$. When considering CD4$^+$ naive T cells as a source, their numbers from age $t_0$ onwards in both facilities were well described with an exponential decay curve, $S(t) = S(t_0)e^{-\nu t}$, though with different exponents (*Figure 1—figure supplement 2A*). With this form, we can solve *Equation (1)* for $M(t)$ explicitly;

$$
\begin{aligned}
M(t) &= M(t_0)e^{\lambda(t_0-t)} + \int_{s=t_0}^{t}\varphi S(s)e^{-\lambda(t-s)}ds \\
&= M(t_0)e^{\lambda(t_0-t)} + \varphi S(t_0)e^{-\lambda t}\int_{s=t_0}^{t}e^{-(\nu-\lambda)s}ds
\end{aligned}
\tag{8}
$$

$$
= M(t_0)e^{\lambda(t_0-t)} + \frac{\varphi S(t_0)}{\lambda-\nu}\left(e^{-\nu t} - e^{\lambda(t_0-t)-\nu t_0}\right).
\tag{9}
$$

Using this expression for $M(t)$ in *Equation (6)*, the kinetics of normalised chimerism in memory are

$$
\begin{aligned}
\frac{d}{dt}\chi_{\text{M,norm}}(t) &= \frac{\varphi S(t_0)e^{-\nu t}}{M(t_0)e^{\lambda(t_0-t)} + \frac{\varphi S(t_0)}{\lambda-\nu}\left(e^{-\nu t} - e^{\lambda(t_0-t)-\nu t_0}\right)} \times \left(\chi_{\text{source,norm}}(t) - \chi_{\text{M,norm}}(t)\right) \\
&= \frac{1}{e^{(\nu-\lambda)t}\left(\frac{e^{-(\nu-\lambda)t_0}}{\nu-\lambda} + \frac{M(t_0)e^{\lambda t_0}}{\varphi S(t_0)}\right) - \frac{1}{\nu-\lambda}} \times \left(\chi_{\text{source,norm}}(t) - \chi_{\text{M,norm}}(t)\right).
\end{aligned}
\tag{10}
$$

The rate of increase in donor chimerism in memory then depends on the force of recruitment $\varphi$, the dynamics and chimerism of the source ($S(t_0)e^{-\nu t}$ and $\chi_{\text{source,norm}}(t)$), the initial memory pool size $M(t_0)$, and the net loss rate $\lambda$. Note that *Equation (10)* predicts that given sufficiently long, and if the rate of decline of naive T cells is less than the rate of loss of memory ($\nu < \lambda$), the chimerism in memory will stabilise at the chimerism of CD4$^+$ naive T cells.

When considering T$_{\text{CM}}$ as a source for T$_{\text{EM}}$ in clean and dirty mice, we described $S(t)$ with sigmoid and exponential decay functions, respectively (*Figure 1—figure supplement 2B*). We described each source's chimerism with the generalised logistic function (*Figure 1—figure supplement 2C and D*).

The homogeneous model is characterised by the three unknowns $M(t_0)$, $\lambda$ and $\varphi$. To estimate them for a given subset, location and source population we solved *Equations (1) and (6)* numerically and fitted them simultaneously to the timecourses of total numbers and normalised chimerism of the memory subset, using a method detailed below. The clean mice underwent BMT at a range of ages, which were accounted for individually in the fitting; model predictions for a mouse which underwent

BMT at age $t_B$ and was observed at age $t$ were generated by running the model from host age $T = t_b + 26d$ (clean) or $t_b + 28d$ (dirty) to time $t$; with $M(T)$ calculated from $M(t_0)$ using *Equation (8)*, and the normalised chimerism at time $T$ assumed to be zero.

## The two-phase model

The two-phase model (*Figure 2B*) describes the kinetics of CD4$^+$ T$_{CM}$ and T$_{EM}$ assuming that both comprise two subpopulations with distinct rates of loss,

$$M(t) = M_{\text{fast}}(t) + M_{\text{slow}}(t), \tag{11}$$

where $t$ is the mouse age. We assume that cells flow only into one subset from the precursor population $S(t)$ at total rate $\varphi S(t)$, and transition to the next compartment at rate $\gamma$;

$$\begin{aligned}
\frac{dM_{\text{fast}}(t)}{dt} &= \varphi S(t) - \gamma M_{\text{fast}}(t) - \lambda_{\text{fast}} M_{\text{fast}}(t) \\
\frac{dM_{\text{slow}}(t)}{dt} &= \gamma M_{\text{fast}}(t) - \lambda_{\text{slow}} M_{\text{slow}}(t).
\end{aligned} \tag{12}$$

Note that despite the nomenclature, when estimating the rates of loss of these subsets, we did not constrain them; rather, the model fits indicated that $\lambda_{\text{fast}} > \lambda_{\text{slow}}$.

Assuming that host and donor cells obey the same kinetics, so that *Equation (12)* hold identically for both populations, then similar to the derivation of *Equation (6)* we obtain the following equations for the dynamics of donor chimerism in the fast and slow subsets, each normalised to the chimerism of DP1 thymocytes;

$$\begin{aligned}
\frac{d}{dt}\chi_{\text{fast,norm}}(t) &= \frac{\varphi S(t)}{M_{\text{fast}}(t)}(\chi_{\text{source,norm}}(t) - \chi_{\text{fast,norm}}(t)) \\
\frac{d}{dt}\chi_{\text{slow,norm}}(t) &= \frac{\gamma M_{\text{fast}}(t)}{M_{\text{slow}}(t)}(\chi_{\text{fast,norm}}(t) - \chi_{\text{slow,norm}}(t)),
\end{aligned} \tag{13}$$

where

$$\chi_{\text{fast,norm}}(t) = \frac{1}{\chi_{\text{DP1}}}\frac{M_{\text{fast}}^{\text{donor}}(t)}{M_{\text{fast}}(t)}, \quad \chi_{\text{slow,norm}}(t) = \frac{1}{\chi_{\text{DP1}}}\frac{M_{\text{slow}}^{\text{donor}}(t)}{M_{\text{slow}}(t)}. \tag{14}$$

The normalised chimerism in the fast and slow populations combined is

$$\chi_{\text{M,norm}}(t) = \chi_{\text{fast,norm}}(t)\frac{M_{\text{fast}}(t)}{M(t)} + \chi_{\text{slow,norm}}(t)\frac{M_{\text{slow}}(t)}{M(t)}. \tag{15}$$

We dealt with different ages at BMT using the same approach described for the simplest model. We determined the initial sizes of the subsets $M_{\text{fast}}(t_0)$ and $M_{\text{slow}}(t_0)$ by assuming that fast cells were in quasi-equilibrium with their source, because all T cell populations change slowly in adult mice (*Figure 1B*); and allowing $M_{\text{slow}}(t_0)$ to be free. Allowing both population sizes to be free yielded very similar results, at the cost of an additional parameter. The numbers of host-derived cells in the fast and slow memory subsets at each time $T$ were then generated from $M_{\text{fast}}(t_0)$ and $M_{\text{slow}}(t_0)$ by running the model forward from age $t_0$ using *Equation (12)*. The two-phase model is then characterised by five unknowns; $M_{\text{slow}}(t_0)$, $\lambda_{\text{fast}}$ and $\lambda_{\text{slow}}$, the transition rate $\gamma$, and the force of recruitment $\varphi$. To estimate these parameters, we fitted the solutions of *Equations (11), (12), and (15)* simultaneously to the timecourses of total memory cell numbers $M(t)$ and the normalised chimerism $\chi_{\text{M,norm}}(t)$, using the empirical forms of $S(t)$ and $\chi_{\text{source,norm}}(\tau)$ where $t$ is host age and $\tau$ is time post-BMT. To visualise the fits to data from the clean facility, we partitioned the mice into four groups based on age at BMT, and plotted the model predictions for the median age at BMT within each group (*Figure 3B*).

## The age-dependent loss model

In this model (*Figure 2C*), the loss rate $\lambda$ is a function of the time since entry of a cell or its ancestor into memory, which we denote its age $a$. The time evolution of the population density of memory cells of age $a$ at host age $t$ is given by

$$\frac{\partial M(t,a)}{\partial t} + \frac{\partial M(t,a)}{\partial a} = -\lambda(a)M(t,a),\tag{16}$$

where the population density of cells of age zero is the rate at which cells flow into memory from the source,

$$M(t, a = 0) = \varphi S(t),\tag{17}$$

and we must specify the overall population density with respect to cell age $M(t_0, a) = g(a)$ at some initial mouse age, $t_0$. As with the other models, we assumed all cells present at $t_0$ are host-derived; we model their age distribution as $g(a) = \varphi S(t_0) e^{pa}$. The free parameter $p$ could be positive or negative, such that older cells can initially be over- or under-represented compared to younger cells. This definition ensures $g(0)$ is the rate of influx of cells of age zero from the source at time $t_0$, $\varphi S(t_0)$. We explored exponential ($\lambda(a) = \lambda_0 e^{-a/A}$) and sigmoid ($\lambda(a) = \lambda_0/(1 + (a/A)^n)$) forms for the dependence of the net loss rate $\lambda$ on cell age, with $n$ = 1, 2, 3, 5 and 10. We found that $n = 3$ consistently yielded the best fits, with the exponential performing the most poorly.

Solving this system using the method of characteristics allows us to track the fates of three cell populations – the host-derived population that was present at $t_0$ ($M_{\text{host}}^{\text{init}}(t,a)$), and host- and donor-derived cells that entered the population after $t_0$ ($M_{\text{host}}^{\text{new}}(t,a)$ and $M_{\text{donor}}(t,a)$). Total memory cell numbers at $t \geq t_0$ are then

$$M_{\text{total}}(t) = \int_{a=0}^{t} \left( M_{\text{host}}^{\text{init}}(t,a) + M_{\text{host}}^{\text{new}}(t,a) + M_{\text{donor}}(t,a) \right) da.\tag{18}$$

The terms in this expression evolve according to

$$\begin{aligned}
M_{\text{host}}^{\text{init}}(t,a) &= g(a - (t - t_0)) \exp\left( -\int_{a-(t-t_0)}^{a} \lambda(s)\, ds \right), \; t - t_0 \leq a \leq t \\
M_{\text{host}}^{\text{new}}(t,a) &= \varphi S_{\text{host}}(t - a) \exp\left( -\int_{0}^{a} \lambda(s)\, ds \right), \; 0 \leq a \leq t - t_0 \\
M_{\text{donor}}(t,a) &= \varphi S_{\text{donor}}(t - a) \exp\left( -\int_{0}^{a} \lambda(s)\, ds \right), \; 0 \leq a \leq t - t_0.
\end{aligned}\tag{19}$$

where $S_{\text{host}}(t) = (1 - \chi_{\text{source}}(t))S(t)$ and $S_{\text{donor}}(t) = \chi_{\text{source}}(t)S(t)$. These expressions give

$$M_{\text{total}}(t) = \int_{a=t-t_0}^{t} g(a-(t-t_0)) \exp\left( -\int_{a-(t-t_0)}^{a} \lambda(s)\, ds \right) da + \varphi \int_{a=0}^{t-t_0} S(t-a) \exp\left( -\int_{0}^{a} \lambda(s)\, ds \right) da.\tag{20}$$

The normalised donor chimerism is

$$\begin{aligned}
\chi_{\text{M,norm}}(t) &= \frac{M_{\text{donor}}(t)}{\chi_{\text{DP1}} M_{\text{total}}(t)} = \frac{\int_{a=0}^{t-t_0} M_{\text{donor}}(t,a)\, da}{\chi_{\text{DP1}} M_{\text{total}}(t)} \\
&= \frac{\varphi}{M_{\text{total}}(t)} \int_{a=0}^{t-t_0} \chi_{\text{source,norm}}(t-a) S(t-a) \exp\left( -\int_{0}^{a} \lambda(s)\, ds \right) da.
\end{aligned}\tag{21}$$

We fitted *Equations (20) and (21)* to the timecourses of their observed counterparts from host age $t_0$ onwards. This model has four free parameters; $p$ and $\varphi$, which, together with the observed value of $S(t_0)$, specify the initial age distribution of host cells, $g(a) = \varphi S(t_0) e^{pa}$ ; and $\lambda_0$ and $A$, which specify the form of $\lambda(a)$. The parameters $p$ and $\varphi$ then determine the initial number of host-derived memory cells;

$$M_{\text{total}}(t_0) \equiv \int_{0}^{t_0} g(a)da = \varphi S(t_0)(e^{pt_0} - 1)/p.\tag{22}$$

As described above, this model was fitted simultaneously to data from mice who underwent BMT at different ages, replacing $t_0$ in *Equation (21)* with the age at BMT plus 26d or 28d for clean and dirty mice, respectively.

## Resistant memory model

In this model, proposed in *Gossel et al. (2017)*, the CD4$^+$ $T_{CM}$ and $T_{EM}$ populations are assumed to be heterogeneous, each consisting of a 'displaceable' subset turning over at rate $\lambda$ and continuously supplemented from the source, and an 'incumbent' or 'resistant' population of host memory cells, $I_{host}(t)$; these are assumed to be established early in life, not supplemented thereafter, and have a distinct net loss rate $\lambda_I$:

$$\frac{dM_{donor}(t)}{dt} = \varphi \chi_{source}(t)S(t) - \lambda M_{donor}(t) \qquad \text{(displaceable, donor)}$$

$$\frac{dM_{host}(t)}{dt} = \varphi(1 - \chi_{source}(t))S(t) - \lambda M_{host}(t) \qquad \text{(displaceable, host)} \qquad (23)$$

$$\frac{dI_{host}(t)}{dt} = -\lambda_I I_{host}(t). \qquad \text{(Incumbent/resistant cells, host)}$$

All donor-derived cells are assumed to be displaceable and obey the same kinetics as displaceable host-derived cells. We solved *Equation (23)* to obtain total memory cell numbers $M_{total}(t) = M_{donor}(t) + M_{host}(t) + I_{host}(t)$, and the normalised chimerism in memory,

$$\chi_{M,norm}(t) = \frac{1}{\chi_{DP1}} \frac{M_{donor}(t)}{M_{total}(t)}. \qquad (24)$$

For simplicity, we assumed that resistant memory cells were stable in number ($\lambda_I = 0$). The resistant memory model then has four free parameters ($M_{total}(t_0)$, $I_{host}(t_0)$, $\varphi$, $\lambda$). Multiple ages at BMT were handled as described for the homogeneous model.

## Parameter estimation and model selection

Each model (with its parameter set denoted $\beta$) was fitted simultaneously to the timecourses of total memory cell numbers $M_i$ and the normalised chimerism $\chi_{norm,i}$, $i = 1 \ldots n$, for a given source population (naive/$T_{CM}$) and environment (dirty/clean). The cell counts and chimerism values were log- and arcsin-square root-transformed, respectively, such that these new variables (denoted $x$ and $y$) could be assumed to have normally distributed errors with constant variances $\sigma_x^2$ and $\sigma_y^2$. We then maximised the joint likelihood of $x$ and $y$ with respect to the model parameters $\beta$, and the unknown $\sigma_x$ and $\sigma_y$. If $X_i(\beta)$ and $Y_i(\beta)$ are the model predictions of the transformed observations $x_i$ and $y_i$ respectively, this likelihood is

$$\mathcal{L} = \prod_i^n \frac{1}{\sqrt{2\pi}\sigma_x} \exp\left(\frac{-(x_i - X_i)^2}{2\sigma_x^2}\right) \times \prod_j^n \frac{1}{\sqrt{2\pi}\sigma_y} \exp\left(\frac{-(x_j - Y_j)^2}{2\sigma_y^2}\right)$$

$$= \frac{\exp(-SSR_x/2\sigma_x^2)}{\left(\sqrt{2\pi}\sigma_x\right)^n} \times \frac{\exp(-SSR_y/2\sigma_y^2)}{\left(\sqrt{2\pi}\sigma_y\right)^n}, \qquad (25)$$

where SSR denotes the sum of squared residuals, and both $SSR_x$ and $SSR_y$ are functions of the parameters $\beta$. The log-likelihood is then

$$\ln \mathcal{L} = -n \ln\left(\sqrt{2\pi}\sigma_x\right) - n \ln\left(\sqrt{2\pi}\sigma_y\right) - \frac{SSR_x}{2\sigma_x^2} - \frac{SSR_y}{2\sigma_y^2}. \qquad (26)$$

To reduce the number of unknowns to be estimated with a parameter search, we substituted the maximum likelihood estimates of the error variances, which can be calculated directly;

$$\frac{\partial \ln \mathcal{L}}{\partial \sigma_x} = -\frac{n}{\sigma_x} + \frac{SSR_x}{\sigma_x^3} = 0 \Longrightarrow \hat{\sigma}_x^2 = \frac{SSR_x}{n},$$

(and similarly for $\hat{\sigma}_y$), giving the following expression for the joint log-likelihood,

$$\ln \mathcal{L} = -\frac{n}{2} \ln\left(SSR_x \times SSR_y\right) - 2n. \qquad (27)$$

This quantity was then maximised with respect to the parameters $\beta$ using the scipy.optimize package in Python. We used the best-fitting model predictions to estimate 95% confidence intervals on

parameters by bootstrapping residuals 1000 times, re-fitting and taking the 2.5% and 97.5% quantiles of the resulting distributions of parameter estimates.

For each model and source, we performed the above procedure separately for data from the clean and dirty environments, and calculated a combined, maximum log likelihood $\ln \mathcal{L}_{\mathrm{combined}} = \ln \mathcal{L}_{\mathrm{clean}} + \ln \mathcal{L}_{\mathrm{dirty}}$. We then used the corrected Akaike Information Criterion, AICc (*Akaike, 1974*; *Burnham and Anderson, 2002*) to assess the relative support for each model/source pairing, where

$$\mathrm{AICc} = -2\ln \mathcal{L}_{\mathrm{combined}} + 2K + \frac{2K(K+1)}{N-K-1}. \tag{28}$$

Here, $K$ is the total number of estimated parameters, which was double the number of parameters in each model (one set for clean and another for dirty); and $N$ is the total number of observations, which was $2 \times$ the number of mice in the clean facility $+ 2 \times$ the number in the dirty facility (each mouse yielded two measurements for each memory subset – total cell numbers, and donor chimerism).

Annotated code and data for performing all analyses are freely available from https://github.com/marianowicka/memory-CD4-and-dirt.git (*Nowicka, 2019*; copy archived at https://github.com/elifesciences-publications/memory-CD4-and-dirt).

## Acknowledgements
This work was supported by the NIH (R01 AI093870) and the Medical Research Council (MC-PC-13055). We thank Fiona Powrie and the Oxford Centre for Microbiome Studies for germ-free mice.

## Additional information

### Funding

| Funder | Grant reference number | Author |
|---|---|---|
| National Institutes of Health | R01 AI093870 | Thea Hogan<br>Maria Nowicka<br>Daniel Cownden<br>Andrew J Yates |
| Medical Research Council | MR/P011225/1 | Benedict Seddon |

The funders had no role in study design, data collection and interpretation, or the decision to submit the work for publication.

### Author contributions
Thea Hogan, Conceptualization, Data curation, Formal analysis, Validation, Visualization, Writing—review and editing; Maria Nowicka, Conceptualization, Data curation, Software, Formal analysis, Validation, Visualization, Writing—review and editing; Daniel Cownden, Software, Formal analysis, Investigation; Claire F Pearson, Resources, Investigation; Andrew J Yates, Conceptualization, Data curation, Software, Formal analysis, Supervision, Funding acquisition, Validation, Investigation, Methodology, Writing—original draft, Project administration, Writing—review and editing; Benedict Seddon, Conceptualization, Data curation, Formal analysis, Supervision, Funding acquisition, Methodology, Writing—original draft, Project administration, Writing—review and editing

### Author ORCIDs
Andrew J Yates (iD) https://orcid.org/0000-0003-4606-4483
Benedict Seddon (iD) https://orcid.org/0000-0003-4352-3373

### Ethics
Animal experimentation: Animal experiments were performed according to the UCL Animal Welfare and Ethical Review Body and Home Office regulations under PPL 70-8310.

**Decision letter and Author response**
Decision letter https://doi.org/10.7554/eLife.48901.sa1
Author response https://doi.org/10.7554/eLife.48901.sa2

## Additional files

### Supplementary files

• Source data 1. All data used in the study.

• Transparent reporting form

### Data availability

The code for mathematical models has been deposited on GitHub (https://github.com/mariano-wicka/memory-CD4-and-dirt; copy archived at https://github.com/elifesciences-publications/memory-CD4-and-dirt), and raw cell counts used for model fitting are provided as source data files, as featured in figures 1, 5 and 7.

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
