## [Decision Letter]

**Acceptance summary:**

In this paper Hogan et al. analyze the development of CD4^+^ T cell memory populations (central and effector) in mice housed under different conditions. They aim to understand the main driver of the establishment of memory population in the absence of infections, and provide insight into relative roles of antigens coming from commensals, self-antigens and pathogens in the establishment and maintenance of the memory T-cell pool throughout life of a mouse. A major finding is that after an early period where commensal / environmental antigens are important in driving memory, the replenishment of new memory has similar dynamics independent of the environment.

**Decision letter after peer review:**

Thank you for submitting your article "Differential impact of self and environmental antigens on the ontogeny and maintenance of CD4^+^ T cell memory" for consideration by *eLife*. Your article has been reviewed by three peer reviewers, including Rob de Boer as the Guest Editor and Reviewer #1, and the evaluation has been overseen by Satyajit Rath as the Senior Editor. The following individuals involved in review of your submission have agreed to reveal their identity: Ruy Ribeiro (Reviewer #2); Jose Borghans (Reviewer #3).

The reviewers have discussed the reviews with one another and the Guest Editor has drafted this decision to help you prepare a revised submission.

Summary:

In this paper Hogan et al. analyze the development of CD4^+^ T cell memory populations (central and effector) in mice housed under different conditions. They aim to understand the main driver of the establishment of memory population in the absence of infections. This is an important and insightful paper that perfectly fits the scope of *eLife*. It is highly original work, very clearly written, and reports important insights into the relative roles of antigens coming from commensals, self-antigens and pathogens in the establishment and maintenance of the memory T-cell pool throughout life (of a mouse). It is also a very timely paper given the recent discussions in the immunological field on the role of exposure to "dirt". The main finding is that after an early period where commensal / environmental antigens are important in driving memory, after a few weeks continuous replenishment of new memory has similar dynamics independent of the environment where the mice are housed. The paper is well-written and clear.

The reviewers were particularly impressed with the novel insights into the maintenance and formation of CD4^+^ memory T cells in mice housed under various conditions.

Essential revisions:

1) The main question is regarding the modeling approach. We appreciate that the authors have tested three different models and chosen the best one. But all the models are linear, and so to achieve the balance observed between source, fast and slow cells requires fine tuning of the dynamical parameters. The question is whether it would make more biological sense to have some kind of feedback or density-dependent mechanism. Although we don't think you have to "invent a more complex model" when the simpler one works well, this could at least be a point of discussion.

2) Another question is that the authors model central and effector memory cells independently. Could these two cell populations be coupled, for example, through interconversion? How would this affect the modeling approach? And more importantly, could it affect the conclusions?

---

## [Author Response]

Essential revisions:1) The main question is regarding the modeling approach. We appreciate that the authors have tested three different models and chosen the best one. But all the models are linear, and so to achieve the balance observed between source, fast and slow cells requires fine tuning of the dynamical parameters. The question is whether it would make more biological sense to have some kind of feedback or density-dependent mechanism. Although we don't think you have to "invent a more complex model" when the simpler one works well, this could at least be a point of discussion.

This is a nice point. We considered it but concluded that a treatment of feedback regulation of cell numbers will not be informative here, for three reasons;

1) The memory cell numbers we observe from age 10 weeks onwards only vary by factor of 2 or 3; the average pool sizes in clean and dirty mice differ by a similar factor (Figure 1B); and we don’t find significant differences in the average net loss rates of each subset in the two environments (Figure 4B and Table 2). Without larger perturbations to cell numbers, then, we do not expect to be able to identify any density dependence of the net loss rates with our data. For the two-compartment model, we are also already at the limit of the number of parameters we are comfortable attempting to estimate.

2) None of the models require fine-tuning to achieve stability. Numbers are simply the balance of slowly declining influx from the naive pool (constant force of recruitment * naive numbers, which decline naturally); and first order decay from one or more sequential compartments. So in these models all memory (sub)compartments can achieve a stable steady state without feedback regulation.

3) Arguably there is little experimental evidence for homeostatic regulation of memory numbers, at least in SPF mice — we see three levels of stable memory in the three facilities, memory pools appears not to fill up rapidly with large clones in very young mice; and memory cell numbers are expandable following multiple infections in older animals (Vezys, Yates et al. Nature 2009).

We have added a discussion of these points.

2) Another question is that the authors model central and effector memory cells independently. Could these two cell populations be coupled, for example, through interconversion? How would this affect the modeling approach? And more importantly, could it affect the conclusions?

In response to other comments we revisited the data with some alternative models (thank you for suggesting this) and in the process improved our descriptor functions for the sources and our strategy for parameter estimation; detailed in Materials and methods. As a result we now clearly favour a naive – CM – EM pathway.

We reached this conclusion by fitting naive -> CM and CM -> EM separately. We were not able to fit all three simultaneously – the number of parameters was too large, and given that we find heterogeneity within both pools there is some uncertainty in how to connect the two models. Even with a constant rate of conversion, solving two coupled models where survival changes continuously with cell age is technically beyond us. Also – since there may be an expansion factor between CM and EM, the two are only coupled weakly as regards fitting.

We did model a Naive-EM-CM pathway but the fits are extremely poor (see Author response image 1). Indeed we argue that such a pathway is inconsistent with the timecourses of chimerism in the two pools. So we choose not to include these fits.

**Author response image 1. respfig1:** Best fit model in which CM cells are fed exclusively by EM cells. The model underestimates the accumulation of CM donor chimerism, as it is constrained by the lower chimerism of the upstream EM cells.